# Explicit Multi-head Attention for Inter-head Interaction in Large Language Models

## Abstract

In large language models built upon the Transformer architecture, recent studies have shown that inter-head interaction can enhance attention performance. Motivated by this, we propose **Multi-head Explicit Attention (MEA)**, a simple yet effective attention variant that explicitly models cross-head interaction. MEA consists of two key components: a Head-level Linear Composition (HLC) module that separately applies learnable linear combinations to the key and value vectors across heads, thereby enabling rich inter-head communication; and a head-level Group Normalization layer that aligns the statistical properties of the recombined heads. MEA shows strong robustness in pretraining, which allows the use of larger learning rates that lead to faster convergence, ultimately resulting in lower validation loss and improved performance across a range of tasks. Furthermore, we explore the parameter efficiency of MEA by reducing the number of attention heads and leveraging HLC to reconstruct them using low-rank "virtual heads". This enables a practical key-value cache compression strategy that reduces KV-cache memory usage by 50% with negligible performance loss on knowledge-intensive and scientific reasoning tasks, and only a 3.59% accuracy drop for Olympiad-level mathematical benchmarks.

## 1 Introduction

Commonly-used attention mechanisms (Vaswani et al., 2023; Shazeer, 2019; Ainslie et al., 2023) treat different heads as independent computational branches, or at most apply simple grouping strategies, without modeling rich cross-head interactions. However, recent studies have demonstrated that incorporating such interactions can significantly enhance the effectiveness of attention mechanisms, through approaches such as leveraging low-rank structures across heads (DeepSeek-AI et al., 2024) or explicitly modeling inter-head communication (Ye et al., 2025; Shazeer et al., 2020).

To better harness the untapped potential of inter-head interactions, we propose **Multi-head Explicit Attention (MEA)**—a simple yet effective attention variant that explicitly strengthens cross-head communication. MEA introduces learnable linear combinations applied separately to the key and value vectors across attention heads, thereby enhancing their mutual interaction. We further show that many existing attention variants are mathematically equivalent to ablated forms of this design. However, such formulations are often prone to degeneration during training, where the intended cross-head communication becomes ineffective. To address this, MEA incorporates GroupNorm (Ye et al., 2025) to stabilize optimization and preserve representational diversity.

In our experiments, we leverage scaling laws to enable efficient hyperparameter selection and validate the design of MEA. Furthermore, we exploit MEA's structure to reduce KV-cache memory usage by 50% with negligible performance degradation.

Our main contributions are summarized as follows:

- We propose Multi-head Explicit Attention (MEA) and introduce the Head-level Linear Composition (HLC) module as its core component.
- We provide a unified theoretical view under the MEA framework, showing that certain ablated variants from Differential Transformer and Talking-Heads Attention can be viewed as special cases within the MEA formulation. We further analyze why MEA provides better optimization behavior compared to these variants.

- By leveraging scaling laws for cost-efficient hyperparameter selection, we conduct from-scratch pretraining experiments and demonstrate that MEA enables faster convergence with larger learning rates, while consistently outperforming baselines.

- MEA naturally supports reducing the number of key-value heads. Leveraging this property, we develop a KV-cache compression strategy that reduces memory usage by 50% with negligible performance degradation.

## 2 PRELIMINARY

To establish a unified framework for analyzing diverse attention mechanisms, we first introduce a consistent set of notations used throughout this paper. Unless otherwise specified, we assume a pre-layer normalization architecture with RMSNorm (Zhang & Sennrich, 2019), and adopt the SwiGLU activation (Shazeer, 2020) in the feedforward networks (FFNs) of all Transformer variants under consideration.

Modern large language models (LLMs) typically employ one of three core attention mechanisms: Multi-Head Attention (MHA)(Vaswani et al., 2023), Multi-Query Attention (MQA)(Shazeer, 2019), and Grouped-Query Attention (GQA) (Ainslie et al., 2023). Among them, GQA serves as a conceptual superset, generalizing both MHA and MQA.

To unify these mechanisms, we adopt a generalized formulation based on a *grouping function* $G :$ $\mathbb{Z}_{[ht]} \to \mathbb{Z}_{[g]}$,[1] which maps each of the $h$ query heads to one of $g$ key-value groups. Under this formulation, each query head $i$ attends to the key-value pair associated with group $G(i)$:

$$\mathbf{Q}_i \in \mathbb{R}^{N \times d_{qk}} = \mathbf{X}\boldsymbol{W}_i^{\mathrm{Q}}, \quad [\mathbf{K}_j, \mathbf{V}_j] \in \mathbb{R}^{N \times (d_{qk}+d_v)} = \mathbf{X}\boldsymbol{W}_j^{\mathrm{KV}}, \tag{1}$$

where $i \in \mathbb{Z}_{[ht]}$ and $j \in \mathbb{Z}_{[g]}$ index the query heads and key-value groups, respectively, and $d_{qk}$, $d_v$ denote the per-head dimensions for queries/keys and values.

Each head then computes its contextual output as:

$$\mathbf{C}_i = \mathrm{softmax}\left(\frac{\phi(\mathbf{Q}_i)\phi(\mathbf{K}_{G(i)})^\top}{\sqrt{d_{qk}}}\right)\mathbf{V}_{G(i)}, \tag{2}$$

where $\phi(\cdot)$ denotes a positional encoding function applied to both queries and keys, such as Rotary Position Embedding (Su et al., 2023).

The outputs of all heads are then concatenated and projected to produce the final attention result:

$$\mathrm{Attention}(\mathbf{X}) = \mathrm{Concat}(\mathbf{C}_1, \ldots, \mathbf{C}_h)\boldsymbol{W}^{\mathrm{O}}. \tag{3}$$

Although $h$ and $g$ differ across MHA, MQA, and GQA, all these mechanisms can be unified under a shared grouping formulation using the function: $G(i) = \left\lceil \frac{i}{h} \cdot g \right\rceil$, which maps each query head $i \in \mathbb{Z}_{[ht]}$ to one of the $g$ key-value groups.

## 3 RELATED WORKS

### 3.1 DIFFERENTIAL TRANSFORMER

Differential Transformer (Ye et al., 2025) observes that attention scores computed via the softmax operation often exhibit substantial noise. This noise reduces the model's ability to focus on truly relevant information, thereby impairing its capacity to retrieve key content from the context—a limitation that becomes particularly pronounced in large language models. To mitigate this issue, Differential Attention (DFA) introduces a mechanism that computes the difference between the attention scores of two distinct heads, effectively canceling out shared noise patterns and enhancing the overall attention quality.

---

[1]Here, $\mathbb{Z}_{[n]}$ denotes the set $\{1, \cdots, n\}$ instead of $\{0, \cdots, n-1\}$ for notational simplicity.

DFA can be interpreted as applying a fixed linear transformation to the attention logits. Specifically, attention heads are grouped into fixed, ordered pairs. For the $i$-th pair $(2i-1, 2i)$, the differential attention logits are computed as:

$$\mathbf{A}_i = \text{softmax}\left(\frac{\phi(\mathbf{Q}_{2i-1})\phi(\mathbf{K}_{2i-1})^\top}{\sqrt{d_{qk}}}\right) - \lambda_i \cdot \text{softmax}\left(\frac{\phi(\mathbf{Q}_{2i})\phi(\mathbf{K}_{2i})^\top}{\sqrt{d_{qk}}}\right), \qquad (4)$$

where $\mathbf{A}_i$ denotes the differential attention score for the $i$-th pair, and $\lambda_i \in \mathbb{R}$ is a learnable scalar for each pair.

Unlike standard attention mechanisms, Differential Attention (DFA) does not treat each head in isolation. Instead, it computes contextual outputs based on joint information from both heads in each pair. Specifically, for the $i$-th head pair, the computation proceeds as:

$$\mathbf{C}_i = \mathbf{A}_i \cdot \text{Concat}\left(\mathbf{V}_{2i-1}, \mathbf{V}_{2i}\right), \qquad (5)$$

$$\text{DFA}(\mathbf{X}) = (1 - \lambda_{\text{init}}) \cdot \text{Concat}\left(\text{GroupNorm}(\mathbf{C}_1, \ldots, \mathbf{C}_{\frac{h}{2}})\right) \boldsymbol{W}^{\text{O}}, \qquad (6)$$

where $\mathbf{A}_i$ denotes the differential attention scores defined in equation 4, and $\mathbf{V}_{2i-1}, \mathbf{V}_{2i}$ are the value vectors for the corresponding heads. The function $\text{GroupNorm}(\cdot)$ represents an RMSNorm operation applied to the concatenated outputs of all head pairs, and $\lambda_{\text{init}}$ is a fixed scalar used to initialize the learnable coefficients $\lambda_i$.

In their paper (Ye et al., 2025), the DFA variant without GroupNorm was shown to perform nearly identically to standard attention. We offer a new perspective to explain this phenomenon: **DFA without GroupNorm can be interpreted as a special case of Talking-Heads Attention that only applies post-softmax transformations**, and its corresponding derivation is provided in Appendix A. Furthermore, we demonstrate that such a formulation tends to degenerate under modern LLM training settings, and analyze the underlying causes of this degeneration. Please refer to Section 4.3 for detailed discussions.

## 3.2 TALKING-HEADS ATTENTION

Talking-Heads Attention (THA) (Shazeer et al., 2020) enhances the expressiveness of MHA by introducing learnable interactions between attention heads. Unlike standard MHA, where each head operates independently and their outputs are merged only at the final stage, THA enables inter-head communication by applying learned linear projections over the attention scores both before and after the softmax operation.

Within our unified framework, THA can be interpreted as introducing two learnable head-level transformation matrices: $\boldsymbol{T}^{\text{QK}} \in \mathbb{R}^{h \times h}$, which is applied to the attention logits before the softmax operation, and $\boldsymbol{T}^{\text{V}} \in \mathbb{R}^{h \times h}$, which is applied to the attention weights after softmax.

Specifically, for each head $i$, the softmax-normalized attention score computation proceeds as:

$$\mathbf{A}_i = \text{softmax}\left(\sum_j \boldsymbol{T}^{\text{QK}}_{i,j} \cdot \frac{\phi(\mathbf{Q}_j)\phi(\mathbf{K}_j)^\top}{\sqrt{d_{qk}}}\right), \qquad (7)$$

where we denote the softmax-normalized attention score map as $\mathbf{A} \in \mathbb{R}^{n \times n \times h}$.

The final context representation and overall attention output are then computed as:

$$\mathbf{C}'_i = \sum_j \boldsymbol{T}^{\text{V}}_{i,j} \cdot \mathbf{A}_j \cdot \mathbf{V}_i, \qquad (8)$$

$$\text{THA}(\mathbf{X}) = \text{Concat}(\mathbf{C}'_1, \ldots, \mathbf{C}'_h)\boldsymbol{W}^{\text{O}}. \qquad (9)$$

By introducing two layers of head-level linear transformation, THA enables richer interactions among attention heads while preserving computational efficiency. However, the original THA design is not compatible with FlashAttention (Dao et al., 2022). In Section 4.3, we will further demonstrate that **under small learning rates, even enhanced THA fails to activate its intended cross-head functionality during training, effectively degenerating into standard MHA behavior**.

## 4 METHOD

In this section, we first introduce a module called the **Head-level Linear Combination (HLC)** module, which serves as a fundamental building block of our approach. We then present our proposed attention mechanism, **Multihead Explicit Attention (MEA)**, in detail. Finally, we present a unified theoretical perspective under the MEA framework, showing that certain ablated variants of Differential Transformer and Talking-Heads Attention can be viewed as incomplete forms of MEA. We further explain why such variants fail to bring consistent improvements from an optimization standpoint.

### 4.1 HEAD-LEVEL LINEAR COMBINATION MODULE

Recent research has explored the idea of matrix decomposition techniques to improve the efficiency and expressiveness of attention mechanisms (DeepSeek-AI et al., 2024; Zhang et al., 2025). Inspired by these developments, we propose a **Head-level Linear Combination (HLC)** module, which reparameterizes the projection tensors by applying a head-level linear combination. This formulation enables parameter sharing and enhanced inter-head interaction without modifying the overall attention structure.

Formally, given a projected head-aware tensor $\mathbf{T} \in \mathbb{R}^{N \times h' \times d}$, we define:

$$\mathrm{HLC}(\boldsymbol{W}_{\mathrm{lc}}^{\mathrm{T}}, \mathbf{T}_{\mathrm{comp}}) := \mathrm{einsum}(\texttt{"n h' d, h' h -> n h d"}, \mathbf{T}, \boldsymbol{W}_{\mathrm{lc}}^{\mathrm{T}}), \tag{10}$$

where $\boldsymbol{W}_{\mathrm{lc}}^{\mathrm{T}} \in \mathbb{R}^{h' \times h}$ is a learnable reweighting matrix that synthesizes $h$ composite heads from $h'$ component attention heads (we assume $h' = h$ unless otherwise specified). These weights are shared across the sequence and feature dimensions, enabling head-level feature mixing while preserving the spatial structure of the input.

### 4.2 MULTIHEAD EXPLICIT ATTENTION (MEA)

In attention modules, query, key, and value vectors are obtained by linearly projecting the hidden activations through learned matrices. The pre-softmax attention scores are computed as inner products between query and key vectors in a shared Hilbert space to form a kernel fuction (Choromanski et al., 2022), often augmented with positional encoding. The final contextual output is computed as a weighted sum over value vectors at the sequence level, where the weights come from a softmax distribution over these inner products. These observations highlight a crucial property: **the representations $q$, $k$, and $v$ exhibit intrinsic linear structure.**

Motivated by this insight, we propose **Multihead Explicit Attention (MEA)**, an attention variant built upon the HLC module. MEA improves the representational flexibility of the attention mechanism by learning explicit, head-level compositions over the key and value tensors prior to attention computation.

Formally, let $\mathbf{K}_{\mathrm{comp}}, \mathbf{V}_{\mathrm{comp}} \in \mathbb{R}^{N \times h' \times d}$ denote the pre-mixed key and value tensors, where $N$ is the sequence length, $h$ is the number of attention heads, and $d$ is the dimensionality per head. MEA applies the following linear transformations:

$$\mathbf{K}_{\mathrm{lc}} = \mathrm{HLC}(\boldsymbol{W}_{\mathrm{lc}}^{\mathrm{K}}, \mathbf{K}_{\mathrm{comp}}), \quad \mathbf{V}_{\mathrm{lc}} = \mathrm{HLC}(\boldsymbol{W}_{\mathrm{lc}}^{\mathrm{V}}, \mathbf{V}_{\mathrm{comp}}), \tag{11}$$

where $\boldsymbol{W}_{\mathrm{lc}}^{\mathrm{K}}, \boldsymbol{W}_{\mathrm{lc}}^{\mathrm{V}} \in \mathbb{R}^{h' \times h}$ are learnable head-combination matrices that project the original $h$ heads into $h'$ mixed heads. By replacing the original key and value tensors $\mathbf{K}$ and $\mathbf{V}$ with their linearly composed counterparts $\mathbf{K}_{\mathrm{lc}}$ and $\mathbf{V}_{\mathrm{lc}}$ in equation 2, MEA preserves the standard attention formulation while enhancing head-level expressiveness.

Inspired by the Differential Transformer, we further stabilize training and align the statistical properties across attention heads by applying Group Normalization over the concatenated head outputs:

$$\mathrm{MEA}(\mathbf{X}) = \mathrm{Concat}(\mathrm{GroupNorm}(\mathbf{C}_1, \ldots, \mathbf{C}_{h'}))\boldsymbol{W}^{\mathrm{O}}, \tag{12}$$

where $\mathrm{GroupNorm}(\cdot)$ denotes an RMSNorm operation applied across the head dimension.

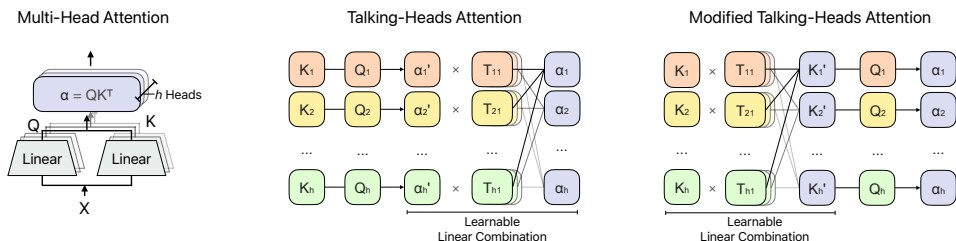

Figure 1: Comparison of how different attention variants compute pre-softmax attention scores, using MHA-based variants as an example. In the modified THA, the learnable linear combination is moved earlier in the computation flow, making it equivalent to MEA's pre-softmax operation.

### 4.3 TOWARDS A UNIFIED VIEW OF ATTENTION MECHANISMS

A widely used positional encoding scheme in LLMs is RoPE (Su et al., 2023), which preserves linear combination under position embedding. Specifically, RoPE fuction $\phi(\cdot)$ satisfies the following property:

$$a_1\phi(\boldsymbol{q}_1) + a_2\phi(\boldsymbol{q}_2) = \phi(a_1\boldsymbol{q}_1 + a_2\boldsymbol{q}_2), \quad b_1\phi(\boldsymbol{k}_1) + b_2\phi(\boldsymbol{k}_2) = \phi(b_1\boldsymbol{k}_1 + b_2\boldsymbol{k}_2), \tag{13}$$

where $a_1, a_2$ and $b_1, b_2$ are arbitrary real scalars.

In equation 7, the learnable linear transformation is applied to the attention scores before the inner product between queries and keys within each head. However, this pre-softmax score mixing scheme, as originally proposed in THA, has been shown to yield negligible performance improvements (Shazeer et al., 2020).

To address this, we move the learnable linear combination module earlier in the computation pipeline (see Figure 1). Generalizing this formulation to the GQA setting and referencing equation 13, we arrive at the following expression:

$$\mathbf{A}_i = \text{softmax}\left(\sum_j \boldsymbol{T}^{\text{QK}}_{G(i),G(j)} \cdot \frac{\phi(\mathbf{Q}_i)\,\phi\left(\mathbf{K}_{G(j)}\right)^\top}{\sqrt{d_{qk}}}\right) \tag{14}$$

$$= \text{softmax}\left(\frac{\phi(\mathbf{Q}_i)\,\phi\left(\sum_j \boldsymbol{T}^{\text{QK}}_{G(i),G(j)} \cdot \mathbf{K}_{G(j)}\right)^\top}{\sqrt{d_{qk}}}\right)$$

$$= \text{softmax}\left(\frac{\phi(\mathbf{Q}_i)\,\phi\left(\text{HLC}(\boldsymbol{T}^{\text{QK}}, \mathbf{K})_{G(i)}\right)^\top}{\sqrt{d_{qk}}}\right), \tag{15}$$

Similarly, we extend the post-attention score mixing formulation in equation 8 to the GQA setting:

$$\mathbf{C}'_i = \sum_j \boldsymbol{T}^{\text{V}}_{G(i),G(j)} \cdot \mathbf{A}_j \cdot \mathbf{V}_{G(i)}, \tag{16}$$

and further modify the computation as:

$$\mathbf{C}'_i = \mathbf{A}_i \cdot \sum_j \boldsymbol{T}^{\text{V}}_{G(i),G(j)} \cdot \mathbf{V}_{G(j)} = \mathbf{A}_i \cdot \text{HLC}(\boldsymbol{T}^{\text{V}}, \mathbf{V})_{G(i)}, \tag{17}$$

where $\text{HLC}(\cdot)$ denotes the Head-level Linear Combination module.

**This reformulation can be shown to be optimization-equivalent to the original post-softmax mixing in THA**; see Appendix B for detailed derivations. Notably, DFA without GroupNorm is also a special case of this post-softmax interaction pattern, and its corresponding derivation is provided in Appendix A.

By comparing with equation 11, we observe that both the modified THA and DFA without Group-Norm adopt the same—or even weaker—pre-attention computation as MEA, applying the HLC

module only to the key-value states. Without additional modifications such as GroupNorm, these formulations tend to degenerate into standard attention, thereby limiting their practical effectiveness.

Consider replacing a standard linear layer weight $\boldsymbol{W}^{\mathrm{T}}$ with an HLC-based reparameterization, denoted as

$$\widetilde{\boldsymbol{W}}^{\mathrm{T}} = \boldsymbol{W}_{\mathrm{lc}} \otimes \boldsymbol{W}_{\mathrm{comp}}, \tag{18}$$

which satisfies that $\widetilde{\boldsymbol{W}}^{\mathrm{T}}\mathbf{X} = \mathrm{HLC}(\boldsymbol{W}_{\mathrm{lc}}, \boldsymbol{W}_{\mathrm{comp}}\mathbf{X})$. Here, the operator $\otimes$ represents a head-wise recombination operation, which can be viewed as a structured generalization of standard matrix multiplication. Importantly, this operator preserves key algebraic properties, including associativity with respect to matrix multiplication and distributivity over matrix addition, thereby maintaining compatibility with gradient-based optimization and compositional transformations. From an optimization perspective, a single gradient update step yields the following parameter changes:

$$\boldsymbol{W}_{t+1}^{\mathrm{T}} = \boldsymbol{W}_t^{\mathrm{T}} + \Delta \boldsymbol{W}_t^{\mathrm{T}}, \tag{19}$$

$$\widetilde{\boldsymbol{W}}_{t+1}^{\mathrm{T}} = \boldsymbol{W}_{\mathrm{lc},t+1}^{\mathrm{T}} \otimes \boldsymbol{W}_{\mathrm{comp},t+1} = (\boldsymbol{W}_{\mathrm{lc},t}^{\mathrm{T}} + \Delta \boldsymbol{W}_{\mathrm{lc},t}^{\mathrm{T}}) \otimes (\boldsymbol{W}_{\mathrm{comp},t} + \Delta \boldsymbol{W}_{\mathrm{comp},t}) \tag{20}$$

$$\approx \boldsymbol{W}_{\mathrm{lc},t}^{\mathrm{T}} \otimes \boldsymbol{W}_{\mathrm{comp},t} + (\Delta \boldsymbol{W}_{\mathrm{lc},t}^{\mathrm{T}} \otimes \boldsymbol{W}_{\mathrm{comp},t} + \boldsymbol{W}_{\mathrm{lc},t}^{\mathrm{T}} \otimes \Delta \boldsymbol{W}_{\mathrm{comp},t}) \tag{21}$$

$$= \widetilde{\boldsymbol{W}}_t^{\mathrm{T}} + \Delta \widetilde{\boldsymbol{W}}_t^{\mathrm{T}}, \tag{22}$$

in the above approximation, we ignore the higher-order interaction term $\mathrm{HLC}(\Delta \boldsymbol{W}_{\mathrm{lc},t}^{\mathrm{T}}, \Delta \boldsymbol{W}_{\mathrm{comp},t})$ under the standard assumption of applying gradient descent in deep learning, where parameter updates are sufficiently small in each step. Consequently, in the absence of non-linear operations—such as GroupNorm—the model is prone to degenerating into standard attention.

**To address this, MEA incorporates Group Normalization inspired by the Differential Transformer, which not only stabilizes training but also helps maintain the expressive cross-head interactions in MEA.**

## 5 EXPERIMENTS

### 5.1 FROM SCRATCH PRE-TRAINING

#### 5.1.1 SETTINGS

**Model**   Our pretraining experiments adopt a model architecture identical to LLaMA3.2-1B (Meta, 2025), except that we do not employ weight tying between the input embedding layer and the output language modeling head. Building upon this base architecture, we explore several attention variants: the original Transformer; a Transformer with GroupNorm applied to head outputs; a MEA variant without GroupNorm (equivalent to the modified THA); the Differential Transformer (Ye et al., 2025); and our proposed Transformer equipped with Multihead Explicit Attention (MEA).

**Training Settings**   We train all models using the AdamW optimizer (Loshchilov & Hutter, 2019) with a weight decay coefficient of $0.1$. The learning rate is annealed to $10\%$ of its initial value using a cosine decay schedule, which promotes smoother convergence in the later stages of training. The default batch size is 20 million tokens, and each model is trained on a total of 500 billion tokens. We follow standard pretraining practices with a curated mixture of text corpora (see Appendix C for details).

#### 5.1.2 COST-EFFICIENT HYPERPARAMETER SELECTION

Established work has shown that increasing network depth often requires proportionally larger learning rates and batch sizes to effectively benefit model performance (D'Angelo et al., 2024; Wang et al., 2022). However, such configurations may also introduce challenges in training stability. Recent studies further suggest that different model architectures demand distinct hyperparameter settings—particularly with respect to learning rate—to achieve optimal training dynamics (Qiu et al., 2025).

Consistent with these findings, we observe that Transformer variants exhibit varying sensitivities to the learning rate, even when the model size is held constant, as we will discuss later in this section.

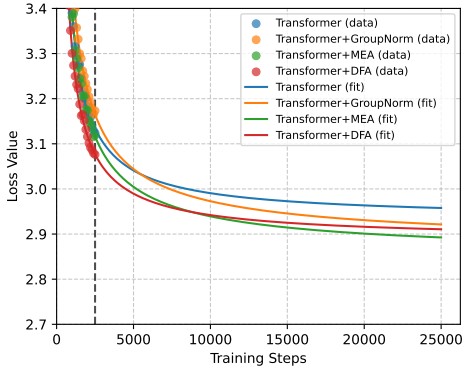 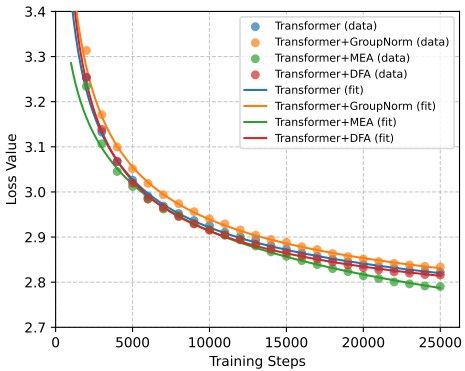

Figure 2: Test loss curves fitted using scaling laws in the learning rate selection experiment, under fixed learning rates $1 \times 10^{-3}$ for different attention variants.

Figure 3: Test loss curves fitted using scaling laws in the full-scale pretraining setting, using cosine decay schedules initialized from the optimal fixed learning rates in Figure 2, comparing different model variants.

Although current pretraining practices often employ relatively small peak learning rates (e.g., $4 \times 10^{-4}$ or $1.5 \times 10^{-4}$) for billion-scale models (Tow et al., 2023; Ye et al., 2025), our results indicate that such choices may not be universally optimal across architectures.

In our experiments, we find that the best-performing learning rate for each variant corresponds to the largest value that avoids unstable behavior such as loss spikes—sharp, unrecoverable increases in loss during training. Models trained under these maximal stable learning rates consistently achieve the most effective reduction in loss on held-out test sets. For further details, please refer to Appendix E.

This observation highlights the necessity for architecture-specific learning rate tuning. However, performing full-scale training for each candidate learning rate is computationally prohibitive. To address this challenge, we adopt the concept of *scaling laws*, which fit a power-law relationship between the loss and the number of training tokens. This allows us to estimate the asymptotic performance of each setting from short-run training trajectories. Further details are provided in Appendix D. This approach enables efficient learning rate selection using only a small fraction of the full pretraining budget.

To this end, we perform a systematic grid search over learning rates for all Transformer variants evaluated in this study. Each variant is trained on 50 billion tokens using four candidate learning rates: $1 \times 10^{-4}$, $5 \times 10^{-4}$, $1 \times 10^{-3}$, and $3 \times 10^{-3}$. We adopt a linear warmup strategy over the first 1000 training steps, followed by standard training with fixed learning rate. All other hyperparameters are kept consistent with the full-scale pretraining configuration.

As shown in Figure 2, MEA consistently achieves lower validation loss under the same learning rate ($1 \times 10^{-3}$) compared to other variants. Moreover, experimental results in Appendix E indicate that MEA-based Transformers maintain stable training behavior at peak learning rates up to $3 \times 10^{-3}$, while other variants become unstable beyond $1 \times 10^{-3}$.

### 5.1.3 FROM SCRATCH PRETRAINING EXPERIMENTS

Building upon the learning rate selection experiments, we identified the maximum stable learning rate for each model variant and adopted a cosine annealing scheduler for full-scale training. We retain 1000 steps for optimizer warm-up. Comparing the test loss curves fitted by scaling laws in Figure 2 and Figure 3, we observe that the cosine scheduler consistently helps the models reach lower final test loss compared to fixed learning rates, while largely preserving the relative ordering predicted by the preliminary experiments.

As illustrated in Figure 3, models with GroupNorm are more difficult to fit during early training, but they exhibit the potential to outperform the vanilla Transformer given sufficient training steps. The behaviors of DFA and MEA are consistent with the trends observed in the learning rate selec-

| Datasets | PIQA | OBQA | WinoGrande | HellaSwag | ARC-e | ARC-c | Avg. |
|----------|------|------|------------|-----------|-------|-------|------|
| Transformer | 71.93 | 21.00 | 56.04 | 40.62 | 59.51 | 26.19 | 45.88 |
| +GroupNorm | 71.38 | 21.00 | **56.12** | 40.59 | 59.13 | 25.77 | 45.67 |
| +DFA | 71.76 | **22.20** | 54.38 | 41.29 | 60.69 | **27.82** | 46.36 |
| Ours | **73.18** | 19.80 | 54.14 | **42.02** | **61.57** | 27.65 | **46.39** |

Table 1: Performance (Accuracy %) on standard NLP benchmarks. **Bold** indicates the best result under the same training configuration.

tion phase: DFA demonstrates faster loss reduction in the early training stages, thereby exhibiting strong early-stage performance. In contrast, MEA effectively mitigates the convergence slowdown typically introduced by GroupNorm, ultimately achieving the best overall performance. These observations are further supported by the downstream evaluation results reported in Table 1. **Notably, the MEA variant without GroupNorm (i.e., the modified version of THA) still exhibits behavior nearly identical to the baseline Transformer, and is thus omitted from the plots and tabular for clarity.**

## 5.2 Compressing Key-Value Cache for Efficient Continued Pretraining

Key-Value cache has long been considered a major bottleneck in the inference stage of LLMs, especially when processing long sequences. In standard autoregressive generation, the attention module of each Transformer layer caches the key and value states computed from all past tokens for use in subsequent causal attention computations. While this mechanism significantly improves inference efficiency, it incurs a memory cost that grows linearly with sequence length $T$ and is further scaled by the number of layers $L$, attention heads $H$, and head dimension $d_k$. Specifically, the space complexity of the KV cache is $\mathcal{O}(LHTd_k)$, posing a substantial demand on memory and becoming a central bottleneck for long-context tasks.

To alleviate this issue, we propose a KV cache compression scheme for the inference stage, inspired by the MEA design. Drawing further inspiration from recent work on SVD-based weight compression (Peng et al., 2025), we apply low-rank approximations to the key and value projection matrices, $W^{\mathrm{K}}$ and $W^{\mathrm{V}}$, yielding the decomposed MEA weight matrices:

$$W^{\mathrm{K}} \approx \widetilde{W}^{\mathrm{K}'} \otimes \widetilde{W}^{\mathrm{K}}_{\mathrm{lc}}, \quad W^{\mathrm{V}} \approx \widetilde{W}^{\mathrm{V}'} \otimes \widetilde{W}^{\mathrm{V}}_{\mathrm{lc}}, \tag{23}$$

where $\widetilde{W}^{\mathrm{K}'}, \widetilde{W}^{\mathrm{K}}_{\mathrm{lc}}, \widetilde{W}^{\mathrm{V}'}, \widetilde{W}^{\mathrm{V}}_{\mathrm{lc}}$ represent the compressed basis and reconstruction matrices for the key and value weights, respectively. Here, the operator $\otimes$ represents a head-wise recombination operation mentioned in equation 18. These matrices are used to **approximate the original multi-head representations using fewer virtual heads**, thereby reducing KV cache memory without modifying the model's forward computation. Please refer to Appendix F for detailed derivations of this approximation scheme.

### 5.2.1 Settings

**Model** We conduct continued pretraining experiments based on the Qwen3-30B-A3B model (Team, 2025). To evaluate the effectiveness of our low-rank approximation strategy described in equation 23, we consider three key-value memory compression configurations with varying granularity. In the *full-layer compression* setting, all Transformer layers adopt MEA-based KV compression. In the *half-layer compression* setting, only layers 12 through 35 (out of 48 in total) are compressed; these layers are selected based on loss sensitivity profiling results detailed in Appendix F, targeting those with minimal impact on validation loss under compression. In the *deep-layer compression* setting, only the first 3 layers remain uncompressed, while all remaining layers apply MEA-based KV compression. **In all compression settings, the number of key-value heads in the compressed layers is reduced from 4 to 2.** For comparison, we also include a *full-parameter continued pretraining* variant without any KV compression as a reference baseline.

**Training settings** All models are trained for one epoch using the AdamW optimizer with a peak learning rate of $1.6 \times 10^{-5}$ and a linear warmup over the first 1000 steps. The batch size is set to 68 million tokens. The learning rate is scheduled via cosine annealing to 0 throughout training, similar

|  | Know. Avg. | Sci. Avg. | Math Avg. | Total Avg. |
|---|---|---|---|---|
| Qwen3-30B-A3B | 63.82 | 48.39 | 65.12 | 58.68 |
| +CPT | 65.81 | 49.76 | 50.48 | 54.39 |
| Half Compression + CPT | 63.54 (-2.27) | 49.44 (-0.32) | 48.49 (-1.99) | 52.94 (-1.45) |
| Deep Compression + CPT | 61.91 (-3.90) | 48.27 (-1.49) | 46.13 (-4.35) | 51.21 (-3.18) |
| Full Compression + CPT | 61.25 (-4.56) | 42.07 (-7.69) | 43.68 (-6.80) | 47.89 (-6.50) |
| Full Compression + Recov. + CPT | 64.41 (-1.40) | 48.79 (-0.97) | 46.89 (-3.59) | 52.36 (-2.03) |

Table 2: Performance (Acc%) on complex reasoning benchmarks. All results are compared against the full-parameter CPT baseline (+CPT), with relative deltas shown in parentheses (smaller is better).

with our main pretraining configuration. We follow standard pretraining practices with a curated mixture of text corpora (see Appendix C for details).

### 5.2.2 ANALYSIS

To assess the impact of memory compression on complex reasoning, we evaluate all models on a suite of challenging benchmarks supported by OpenCompass. These tasks span diverse reasoning categories: MMLU-Pro (Liu et al., 2024a), GPQA Diamond (Rein et al., 2023), and SuperG-PQA (Liu et al., 2024c) for *knowledge reasoning*; ChemBench (Chen et al., 2024), ClimaQA (Wei et al., 2024), and MedXpertQA (Wang et al., 2024) for *scientific reasoning*; and AIME 2025 (Contributors, 2023), OlympiadBench (Liu et al., 2024b), LiveMathBench-Hard (Sun et al., 2024), and OlymMATH (Sun et al., 2025) for *mathematical reasoning*. The summarized results are presented in Table 2, and a detailed breakdown of sub-task scores is provided in Appendix G.

Experimental results show that **even under full compression, the model can recover to a competitive performance level when continued pretraining (CPT) is combined with an additional recovery stage**. Furthermore, knowledge and science reasoning tasks appear relatively robust to this form of memory compression, with minimal performance degradation. In contrast, mathematical reasoning is more sensitive: even the full-parameter CPT baseline exhibits a noticeable performance drop, likely due to differences in data quality compared to the original Qwen3 pretraining. Nevertheless, our fully compressed variant with Recover+CPT manages to regain acceptable performance across most math tasks. **We believe this performance gap could be further narrowed with improved training data, especially considering that the full-parameter CPT setting also suffers a significant drop in math reasoning.**

## 6 CONCLUSION

In this work, we revisit the role of inter-head interactions in attention variants used in Transformer and propose Multi-head Explicit Attention (MEA) as a simple yet effective extension. By applying learnable linear combinations across attention heads and stabilizing training with GroupNorm, MEA enhances cross-head communication while preserving model robustness. We also show that MEA provides a unified view of several existing variants and supports key-value compression for memory efficiency.

We empirically validate MEA through a series of from-scratch pretraining experiments. First, by leveraging scaling laws, we efficiently select learning rates without exhaustive sweeps, and find that MEA tolerates significantly larger learning rates than baselines. This property allows MEA to converge faster during training, while consistently achieving stronger performance. Second, we exploit MEA's structure to compress key-value states by reducing the number of KV heads. This leads to a 50% reduction in KV-cache memory usage, while maintaining comparable performance on knowledge-intensive and scientific reasoning tasks, and incurring only minor degradation on the most challenging mathematical benchmarks.

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

## A    DFA WITHOUT GROUPNORM

In Section 3.1, we discussed that DFA without GroupNorm can be viewed as a special case of Talking-Heads Attention (THA) that only applies post-softmax transformations. While this connection may not be immediately obvious, we provide further clarification below.

By removing GroupNorm, we effectively eliminate the rescaling factor $1 - \lambda_{\text{init}}$ associated with the normalization term (Ye et al., 2025). Consequently, DFA degenerates into a form where attention outputs are modified exclusively through post-softmax head mixing—precisely aligning with the structure of THA that operates only on attention weights after softmax:

$$\mathbf{A}_i = \text{softmax}\left(\frac{\phi(\mathbf{Q}_{2i-1})\phi(\mathbf{K}_{2i-1})^\top}{\sqrt{d_{qk}}}\right) - \lambda_i \cdot \text{softmax}\left(\frac{\phi(\mathbf{Q}_{2i})\phi(\mathbf{K}_{2i})^\top}{\sqrt{d_{qk}}}\right), \quad (24)$$

$$\mathbf{C}_i = \mathbf{A}_i \cdot \text{Concat}\left(\mathbf{V}_{2i-1}, \mathbf{V}_{2i}\right), \quad (25)$$

$$\text{DFA}(\mathbf{X}) = \text{Concat}\left(\mathbf{C}_1, \ldots, \mathbf{C}_{\frac{h}{2}}\right)\boldsymbol{W}^{\text{O}}, \quad (26)$$

In this setting, we can construct an equivalent THA-style transformation matrix:

$$T_{i,j}^{\text{V}} = \begin{cases} 1, & i \in \{2d-1, 2d\}, \ j = 2d-1, \\ -\lambda, & i \in \{2d-1, 2d\}, \ j = 2d, \\ 0, & \text{otherwise}, \end{cases} \quad (27)$$

where $d \in \mathbb{Z}_{[\frac{h}{2}]}$ denotes the $d$-th head pair in DFA.

Thus, it becomes evident that DFA without GroupNorm is essentially a special case of THA that only applies a post-softmax combination of attention heads.

## B    MODIFICATION ON POST-SOFTMAX MIXING OF THA

In Section 4.3, we move the linear transferring module from acting on the attention scores to acting after the value combination $\mathbf{V}$. In this section, we examine whether there exists any difference in representational capacity between these two formulations.

The original THA formulation computes:

$$\mathbf{C}_i' = \sum_j \boldsymbol{T}_{G(i),G(j)}^{\text{V}} \cdot \mathbf{A}_j \cdot \mathbf{V}_{G(i)} \quad (28)$$

In the modified version:

$$\mathbf{C}_i' = \mathbf{A}_i \cdot \sum_j \boldsymbol{T}_{G(i),G(j)}^{\text{V}} \cdot \mathbf{V}_{G(j)} \quad (29)$$

The final attention output is then:

$$\text{THA}(\mathbf{X}) = \text{Concat}(\mathbf{C}_1', \ldots, \mathbf{C}_h')\boldsymbol{W}^{\text{O}} \quad (30)$$

To analyze the difference, consider the output of the attention layer $\mathbf{Y} = \text{THA}(\mathbf{X})$ at position $p$, denoted as $\boldsymbol{y} = \mathbf{Y}[p]$. We examine the value of a single channel $ch$, written as $y = \boldsymbol{y}[ch]$. Expanding the linear projection:

$$y = \sum_i \sum_k \boldsymbol{c}_i'[k] \cdot \boldsymbol{W}^{\text{O}}[i \cdot \text{head\_dim} + k, ch] = \sum_i \sum_k \boldsymbol{c}_i'[k] \cdot w_{i,k} \quad (31)$$

where $w_{i,k} := \boldsymbol{W}^{\text{O}}[i \cdot \text{head\_dim} + k, ch]$ is the scalar weight applied to the $k$-th dimension of the $i$-th head output when computing the $ch$-th output channel.

Under the original THA formulation:

$$y_{\text{THA}} = \sum_{i,j,k} \boldsymbol{T}_{G(i),G(j)}^{\text{V}} \cdot \boldsymbol{a}_j \cdot \mathbf{V}_{G(i)}[k] \cdot w_{i,k} \quad (32)$$

where $\boldsymbol{a}_j = \mathbf{A}_j[p]$ denotes the attention score (typically from softmax) for position $p$ to query on keys from position $[1, \cdots, p]$, and $\mathbf{V}_{G(i)}[k]$ denotes the $k$-th component of the value vectors from head $G(i)$.

Under the modified formulation:

$$y_{\text{THA}'} = \sum_{i,j,k} \boldsymbol{a}_i \cdot \boldsymbol{T}^{\text{V}}_{G(i),G(j)} \cdot \mathbf{V}_{G(j)}[k] \cdot w_{i,k} \tag{33}$$

where $\boldsymbol{a}_i = \mathbf{A}_i[p]$ denotes the scalar attention score (typically from softmax) for position $p$, and $\mathbf{V}_{G(i)}[k]$ denotes the $k$-th component of the value vectors from head $G(i)$.

From an optimization and representational perspective, these two forms are equivalent in expressive power: the change in summation order can be compensated by adjustments in the learnable transfer matrix $\boldsymbol{T}^{\text{V}}$ and the output projection $\boldsymbol{W}^{\text{O}}$. Therefore, both variants are expected to converge to similarly expressive solutions (having same weights) during training.

## C    Training Data

**From Scratch Pre-training**    We construct pre-training corpus by extending several publicly available datasets, including Common Crawl, FineWeb-Edu (Lozhkov et al., 2024), RefinedWeb (Penedo et al., 2023), StarCoder (Li et al., 2023), and the Stack (Kocetkov et al., 2023). Additionally, we collect supplementary text data from the open web to further improve the diversity of the training set. Following prior practices (Qiu et al., 2024), we apply a five-stage preprocessing pipeline comprising: (1) language identification and text extraction, (2) heuristic rule-based cleaning, (3) fuzzy deduplication, (4) safety filtering, and (5) quality-based data selection. The validation set is selected from Pile-CC, with the objective of aligning test loss behavior with the relative performance rankings observed on downstream tasks.

**Compressing Key-Value Cache for Efficient Continued Pretraining**    The dataset used for continued pretraining is the same as described above. For the full-layer compression setting, we include an additional stage of 1T-token pretraining to recover distributional alignment. For all other settings, training is conducted on a high-quality filtered subset of the original corpus, selected based on data quality scores (Qiu et al., 2024). This high-quality dataset contains approximately 22 billion tokens.

## D    Scaling Laws

Researchers at OpenAI have shown that during model training, the test loss exhibits a predictable power-law relationship with respect to model size and the amount of training data (Kaplan et al., 2020). The most general form of this empirical law can be expressed as:

$$L(N, D) = \left[ \left( \frac{N_c}{N} \right)^{\frac{\alpha_N}{\alpha_D}} + \frac{D_c}{D} \right]^{\alpha_D}, \tag{34}$$

where $L(N, D)$ estimates the test loss given model size $N$ (excluding embeddings) and number of training tokens $D$. The constants $N_c$, $D_c$, $\alpha_N$, and $\alpha_D$ depend on the model architecture, dataset, and evaluation distribution, and are typically estimated via curve fitting.

For a fixed model architecture (i.e., fixed $N$), the above formulation simplifies to a function of data scaling only. By performing a Taylor expansion and omitting lower-order terms, a more practical form is commonly used:

$$L(D) = \left( \frac{D_c}{D} \right)^{\alpha_D} + L_0, \tag{35}$$

where $L_0$ denotes the irreducible loss floor after convergence, and $D_c$, $\alpha_D$ are task- and model-dependent constants. Unless otherwise specified, all references to "scaling laws" in this paper refer to the simplified formulation in Equation 35.

# E  LEARNING RATE SELECTION

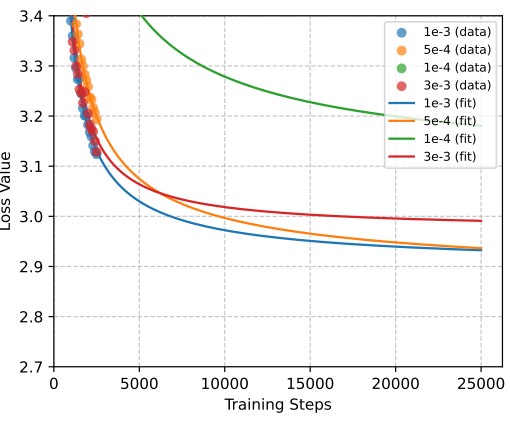

Figure 4: Test loss curves fitted using scaling laws in the learning rate selection experiment, under different fixed learning rates for Transformer.

Figure 5: Test loss curves fitted using scaling laws in the learning rate selection experiment, under different fixed learning rates for Transformer+GroupNorm.

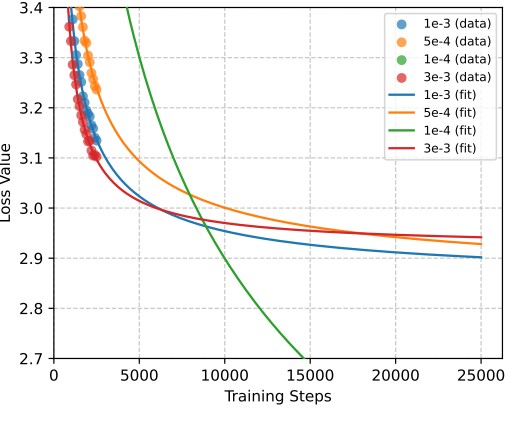

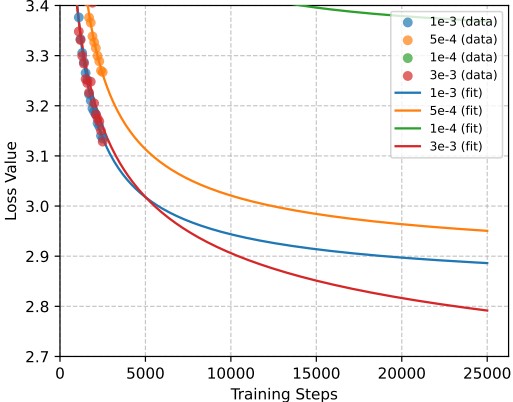

Figure 6: Test loss curves fitted using scaling laws in the learning rate selection experiment, under different fixed learning rates for Transformer+DFA.

Figure 7: Test loss curves fitted using scaling laws in the learning rate selection experiment, under different fixed learning rates for Transformer+MEA.

For the $1 \times 10^{-4}$ learning rate setting, we observe abnormal fitting behavior in Figure 5 and Figure 6, where the empirical loss curve significantly deviates from the expected trend modeled by the scaling law formulation. This is likely due to insufficient training signal under such a small learning rate. While the fit remains formally feasible, the trend is substantially slower than other configurations, making it unlikely to catch up within the 25,000-step training budget.

# F MEA-BASED DERIVATION FOR KEY-VALUE COMPRESSION

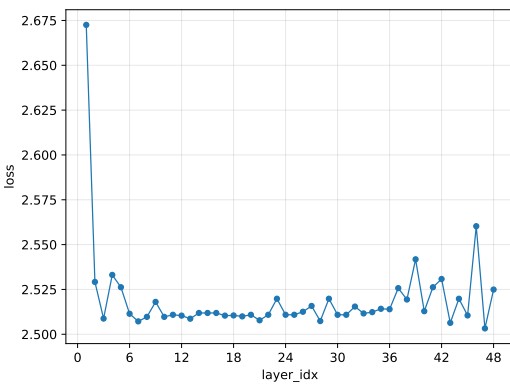

Figure 8: Cross-entropy loss changes when compressing different layers of the key-value cache. The x-axis denotes the index of the compressed layer (ordered from embedding-side to LM head, ranging from 1 to 48), and the y-axis shows the cross-entropy loss measured on a Pile-CC validation subset.

Our compression method is grounded in the observation that considerable redundancy exists among attention heads during generation, particularly in large language models (LLMs). Many heads produce similar representations or exhibit diminished functional diversity. Exploiting this, we apply Singular Value Decomposition (SVD) and low-rank approximation to project the multi-head key and value representations of each token into a more compact subspace—**effectively reducing the number of heads that must be cached**.

Specifically, MEA introduces a linear combination matrix to approximate multiple attention heads using a smaller number of "virtual heads." Consider the key projection matrix $\boldsymbol{W}^{\mathrm{K}} \in \mathbb{R}^{\mathrm{Dim} \times (H \cdot d_k)}$, where Dim is the input embedding dimension and $d_k = \mathrm{Dim}/H$ is the per-head dimensionality. We reshape this matrix as follows:

$$\boldsymbol{W}^{\mathrm{K}} \in \mathbb{R}^{(d_k \cdot \mathrm{Dim}) \times H}, \tag{36}$$

where each column corresponds to a flattened projection for a single head. Applying SVD yields:

$$\boldsymbol{W}^{\mathrm{K}} = \boldsymbol{W}^{\mathrm{K}'} \cdot \Lambda \cdot \boldsymbol{W}^{\mathrm{K}}_{\mathrm{lc}}, \tag{37}$$

where:

- $\boldsymbol{W}^{\mathrm{K}'} \in \mathbb{R}^{(d_k \cdot \mathrm{Dim}) \times H'}$ contains the left singular vectors (basis);
- $\Lambda \in \mathbb{R}^{H' \times H'}$ is a diagonal matrix of singular values;
- $\boldsymbol{W}^{\mathrm{K}}_{\mathrm{lc}} \in \mathbb{R}^{H' \times H}$ is the linear combination matrix.

In typical settings where $\mathrm{Dim} \gg H$, the decomposition is full-rank. By retaining only the top-$H'$ singular values and incorporating $\Lambda$ into $\boldsymbol{W}^{\mathrm{K}}_{\mathrm{lc}}$, we obtain the low-rank approximation:

$$\boldsymbol{W}^{\mathrm{K}} \approx \widetilde{\boldsymbol{W}}^{\mathrm{K}'} \otimes \widetilde{\boldsymbol{W}}^{\mathrm{K}}_{\mathrm{lc}}, \tag{38}$$

where both $\widetilde{\boldsymbol{W}}^{\mathrm{K}'}$ and $\widetilde{\boldsymbol{W}}^{\mathrm{K}}_{\mathrm{lc}}$ are significantly smaller than the original matrix. Here, the operator $\otimes$ represents a head-wise recombination operation mentioned in equation 18.

This procedure can be similarly applied to the value projection matrix $\boldsymbol{W}^{\mathrm{V}}$, enabling compression of both key and value components in the attention cache. Crucially, this approximation leaves the model's original parameters untouched and can be deployed purely at inference time via lightweight linear mappings, offering strong deployment compatibility and minimal computational overhead.

To design layer-wise key-value cache compression strategies, we first perform a layer selection study on the Pile-CC validation set. Specifically, we apply head compression to one layer at a time and record the resulting cross-entropy loss, as illustrated in Figure 8. The x-axis represents the index

of the compressed layer, while the y-axis shows the corresponding cross-entropy loss measured on a subset of Pile-CC. This probing experiment guides the design of our partial-layer compression strategies. The compression method follows the formulation in equation 23. Notably, we observe that compressing the middle layers yields negligible degradation in validation loss compared to compressing early or late layers. Based on these findings, we select layers 12 through 35 (hf model index 11–34) for the half-layer compression configuration in our continued pretraining experiments.

# G  DOWNSTREAM TASK SCORE FOR CPT

| | MMLU-Pro | GPQA Diamond | SuperGPQA | Know. Avg. |
|---|---|---|---|---|
| Qwen3-30B-A3B | 77.52 | 61.62 | 52.32 | 63.82 |
| +CPT | 80.71 | 64.65 | 52.08 | 65.81 |
| Half Compression+CPT | 77.77 | 63.13 | 49.73 | 63.54 |
| Deep Compression+CPT | 77.10 | 61.87 | 46.75 | 61.91 |
| Full Compression+CPT | 77.28 | 60.10 | 46.36 | 61.25 |
| Full Compression+Recov.+CPT | 78.28 | 66.16 | 48.79 | 64.41 |

Table 3: Performance (Acc%) on knowledge reasoning benchmarks.

| | PHYSICS | ChemBench | ClimaQA | MedXpertQA | Sci. Avg. |
|---|---|---|---|---|---|
| Qwen3-30B-A3B | 37.74 | 66.12 | 64.79 | 24.90 | 48.39 |
| +CPT | 36.56 | 70.02 | 65.36 | 27.10 | 49.76 |
| Half Compression+CPT | 37.18 | 71.74 | 63.42 | 25.43 | 49.44 |
| Deep Compression+CPT | 34.67 | 69.93 | 62.17 | 26.29 | 48.27 |
| Full Compression+CPT | 32.14 | 54.19 | 59.33 | 22.61 | 42.07 |
| Full Compression+Recov.+CPT | 35.10 | 72.02 | 63.20 | 24.82 | 48.79 |

Table 4: Performance (Acc%) on scientific reasoning benchmarks.

| | AIME 2025 | OlympiadBench | LiveMathBench-hard | OlymMATH | Math Avg. |
|---|---|---|---|---|---|
| Qwen3-30B-A3B | 80.00 | 73.40 | 55.56 | 51.50 | 65.12 |
| +CPT | 56.67 | 69.77 | 42.22 | 33.25 | 50.48 |
| Half Compression+CPT | 46.67 | 66.11 | 46.67 | 34.50 | 48.49 |
| Deep Compression+CPT | 48.75 | 64.10 | 43.59 | 28.06 | 46.13 |
| Full Compression+CPT | 40.00 | 64.54 | 44.44 | 25.75 | 43.68 |
| Full Compression+Recov.+CPT | 46.67 | 66.91 | 42.22 | 31.75 | 46.89 |

Table 5: Performance (Acc%) on mathematical reasoning benchmarks.

