# OpenReview forum: "Explicit Multi-head Attention for Inter-head Interaction in Large Language Models"
_ICLR.cc/2026/Conference — ICLR 2026 Conference Desk Rejected Submission_

### Official Review · Reviewer_HPL3 · 2025-10-30

**Soundness:** 2
**Presentation:** 2
**Contribution:** 2
**Rating:** 4
**Confidence:** 2

**Summary:**

This study revisits inter-head interactions in Transformer attention blocks by proposing Multihead Explicit Attention (MEA) that leverages a learnable reweighting of head-level KV compositions prior to the attention computation. Crucially, MEA employs GroupNorm over the concatenated head outputs and the work provides theoretical arguments for MEA's formulation. Pretraining experiments demostrate improved performance with faster convergence, with the added benefit of possible compute saving via a reduced number of key-value heads.

**Strengths:**

- Presents useful efforts to unify existing formulations, notably expressing variant of the Differential Transformer and Talking-Heads as special cases of MEA. Specifically, the work offers a new perspective on a DFA variant result, connecting insights across related work.
- Results show improved pretraining convergence and performance with accuracy improvements for some standard benchmark datasets.

**Weaknesses:**

The key premise of the work is that inter-head interaction can enhance attention performance but the experimental results suggest that group normalization may play a bigger role than the mere inter-head communication. Since there are multiple somewhat orthogonal explorations going on, namely an assessment of MEA performance, an exploration of SVD-based efficiency improvements, as well as cost-efficient hyperparameter selection via scaling laws, the main empirical results do not seem to offer much insight into the role of inter-head interaction. The manuscript could benefit from a clearer through-line with an explicit focus on delineating the role inter-head mixing vs. normalization. For example, section 4.2 brings up the Differential Transformer and the role of GroupNorm. However, the argument for this step is only discussed at the end of 4.3, repeating parts of 4.2. Importantly, the introduced weight update analysis is not used in any empirical analysis. A clearer focus and greater efforts in disentangling different effects empirically could greatly strengthen the work.

**Questions:**

- Do you have any insight as to why the pretraining performance gains do not lead to improvements for OBQA WinoGrande and ARC-c?
- Are you planning to release your source code?
- Minor: Throughout the manuscript, sentences are highlighted in bold, presumably to highlighting key results. However, other highlights such as L293 or L391 do not seem to follow this pattern. What are the highlights supposed to achieve?

---

> ### Author Response · Authors · 2025-11-25
> **Response to Reviewer HPL3**
>
> # Response to W1: Role of Group Normalization vs. Inter-Head Communication
>
> Since the introduction of the Differential Transformer, we have recognized that GroupNorm plays an important role in its architecture. However, consistent with the ablation results reported in the Differential Transformer paper, simply applying a variant of GroupNorm alone does not lead to significant performance improvements. Through our study, we observed that incorporating additional normalization can help the model converge to better optima, but it also tends to slow down early-stage convergence. To address this, we drew inspiration from the reparameterization schemes for $\lambda$ in the Differential Transformer, as well as early-stage head level attention score scaling experiments based on muP (arXiv:2203.03466). Our findings suggest that a reparameterization, when combined with information sharing at the attention head level, can significantly accelerate early convergence.
> By revisiting the problem from an optimization perspective and analyzing why this specific combination avoids degeneration, we were able to identify the key insight that supports our simple yet effective design. Although the concept of inter-head interaction is not entirely new, this does not diminish the contribution of our study.
>
> # Response to W2: Multiple Orthogonal Directions in the Paper
>
> The central theme of our paper is the design of an attention alternative that maintains a balanced performance across training efficiency, inference efficiency, and downstream capability. While each direction can certainly be explored further in future work, we believe that achieving this balance already represents meaningful progress for an attention replacement.
>
> # Response to W3: Clarifying the Roles of Inter-Head Mixing and Normalization
>
> We appreciate the suggestion. We have considered presenting Section 3.2 (Talking-Heads Attention, TFA) before Section 3.1 (Differential Transformer, DFA), since DFA is a special case of TFA. Our current structure follows a “special case–core mechanism–generalization” narrative for clarity. We will take this feedback into account and may revise the ordering in future versions to further highlight the distinction between mixing and normalization.
>
> # Response to W4: Use of Weight Update Analysis in Experiments
>
> This analysis is indeed used in our empirical results. In lines 391–393, we emphasize that:
> **Notably, the MEA variant without GroupNorm (i.e., the modified version of THA) still exhibits behavior nearly identical to the baseline Transformer, and is thus omitted from the plots and tables for clarity.**
> This corresponds to the ablation of applying only HLC (also remarked as the modified version of THA). Across downstream tasks, its performance is indistinguishable from the unmodified baseline, so we removed it from the figures for readability. This serves as the empirical counterpart of the analysis presented earlier.
>
> # Response to Q1: Lack of Gains on OBQA, WinoGrande, and ARC-c
>
> For 1B models, these tasks are particularly challenging. Similar observations have been reported even for larger models, such as those in the [NeurIPS 2025 Spotlight] TPA: Tensor ProducT ATTenTion Transformer (T6) (https://arxiv.org/abs/2501.06425).
>
>
> # Response to Q2: Source Code Release Plan
>
> Yes, the source code will be released once the paper is published.
>
> # Response to Q3: Purpose of Bolded Sentences in the Manuscript
>
> Line 293 highlights:
> **To address this, MEA incorporates Group Normalization inspired by the Differential Transformer, which not only stabilizes training but also helps maintain the expressive cross-head interactions in MEA.**
> This aligns with our explanation in the response to W1 regarding the complementary roles of GroupNorm and HLC.
>
>
> In lines 391–393, we emphasize that:
> **Notably, the MEA variant without GroupNorm (i.e., the modified version of THA) still exhibits behavior nearly identical to the baseline Transformer, and is thus omitted from the plots and tables for clarity.**
> This corresponds to the ablation of applying only HLC. Across downstream tasks, its performance is indistinguishable from the unmodified baseline, so we removed it from the figures for readability. This serves as the empirical counterpart of the analysis presented earlier.

---

> > ### Comment · Reviewer_HPL3 · 2025-11-25
> >
> > Thank you for your clarifications; I have no further questions.

---

### Official Review · Reviewer_mviB · 2025-10-30

**Soundness:** 2
**Presentation:** 3
**Contribution:** 2
**Rating:** 4
**Confidence:** 4

**Summary:**

This paper proposes Multi-head Explicit Attention (MEA), a novel attention mechanism designed to improve upon standard Multi-Head Attention (MHA) by explicitly modeling inter-head interaction. The authors identify that standard MHA and its variants treat heads independently, limiting their potential. MEA introduces two key components to address this: 1) a Head-level Linear Composition (HLC) module, which applies learnable linear combinations across heads to the Key (K) and Value (V) vectors before the attention computation, and 2) a head-level Group Normalization layer applied after the attention computation to stabilize training and prevent the model from degenerating.

**Strengths:**

1. The most significant contribution of this work is the KV-cache compression strategy. A 50% reduction in KV-cache memory  is a highly valuable engineering result, directly addressing one of the primary bottlenecks in long-context LLM inference. The fact that this is achieved with "negligible performance loss" on knowledge and science benchmarks (e.g., <1.5% drop on average) is very compelling.

2. The paper provides a clear and convincing argument for why MEA works. The authors demonstrate that both components are necessary. They show that an MEA variant without GroupNorm (which they frame as a modified Talking-Heads Attention) fails to activate cross-head communication and degenerates, performing almost identically to the baseline Transformer.

**Weaknesses:**

1. The idea of inter-head interaction is not new, as the authors acknowledge by citing Talking-Heads Attention and Differential Transformer. The HLC module is a specific form of linear combination, and the GroupNorm component is directly inspired by DFA. The primary innovation is the specific combination of these ideas (pre-attention K/V mixing + post-attention GroupNorm) and the analysis of why this specific combination avoids the degeneration that plagued prior work.
2. While the training dynamics (loss, convergence speed) are improved, the final downstream performance gains from pre-training are very small. In Table 1, the MEA model achieves an average of 46.39%, which is only a minor improvement over the baseline Transformer's 45.88% and DFA's 46.36%.
3. The KV-cache compression is not a "free" operation that can be applied to any model. The experiments (Section 5.2) apply this compression to a pre-trained 30B model and then require a "continued pretraining (CPT)" stage , and in the best-performing case, an additional "recovery stage", to regain performance. This extra training cost is a key part of the trade-off and should be considered when evaluating the overall efficiency.
4. The paper correctly cites the DeepSeek paper, but it stops short of a direct comparison. DeepSeek's "Multi-head Latent Attention" (MLA) is a major concurrent method that tackles the exact same problem (KV cache bottleneck).

**Questions:**

What is the computational and time cost of the "Continued Pretraining (CPT)" and "Recovery" stages used in the 30B model compression experiments? A clearer picture of this cost is needed to fully evaluate the trade-off against the inference savings.

Bedies, the pre-training performance gains on downstream tasks were very modest. Do the authors have evidence or a hypothesis that these gains would become more significant at a larger model scale (e.g., 70B+), or is the primary benefit of MEA truly limited to training/inference efficiency?

---

> ### Author Response · Authors · 2025-11-25
> **Response to Reviewer mviB**
>
> # Response to W1: Novelty of Inter-Head Interaction
>
> Since the introduction of the Differential Transformer, we have recognized that GroupNorm plays an important role in its architecture. However, consistent with the ablation results reported in the Differential Transformer paper, simply applying a variant of GroupNorm alone does not lead to significant performance improvements. Through our study, we observed that incorporating additional normalization can help the model converge to better optima, but it also tends to slow down early-stage convergence. To address this, we drew inspiration from the reparameterization schemes for $\lambda$ in the Differential Transformer, as well as early-stage head level attention score scaling experiments based on muP (arXiv:2203.03466). Our findings suggest that a reparameterization, when combined with information sharing at the attention head level, can significantly accelerate early convergence.
> By revisiting the problem from an optimization perspective and analyzing why this specific combination avoids degeneration, we were able to identify the key insight that supports our simple yet effective design. Although the concept of inter-head interaction is not entirely new, this does not diminish the contribution of our study.
>
>
> # Response to W2: Marginal Final Downstream Gains
>
> Our findings show that even without explicit denoising of attention scores, enabling attention scores to be freely combined with HLC and Group Normalization achieves performance comparable to the Differential Transformer. This outcome demonstrates the validity of our approach and provides valuable insight for future research directions. Furthermore, it is typical for architectural innovations to yield only moderate improvements in downstream performance. For reference, see [NeurIPS 2025 Spotlight] TPA: Tensor ProducT ATTenTion Transformer (T6) (https://arxiv.org/abs/2501.06425).
>
>
> # Response to W3: Applicability and Cost of KV-Cache Compression
>
> It is true that KV-cache compression is not entirely "free" and may require additional steps such as continued pretraining (CPT) or a recovery stage, as applied to the 30B model in Section 5.2. However, it is important to note that if full-layer compression is not required, as in our Deep Compression scheme, retaining 53.125% of the KV cache memory can still achieve strong performance. In this case, applying CPT over 22B tokens represents a modest computational overhead. Below is a table summarizing the remaining KV ratio for each compression level:
>
>
> | Model Compression Stage | Remained KV Ratio |
> |------------------------|-------------------|
> | Half Compression       | 75%               |
> | Deep Compression       | 53.125%           |
> | Full Compression       | 50%               |
>
>
> # Response to W4: Lack of Direct Comparison with DeepSeek MLA
>
> In fact, we have experimented with the MHA2MLA approach (https://arxiv.org/abs/2502.14837). From the perspective of equivalent computational cost, its scalability is suboptimal. Additionally, due to the design of Qwen models, such as qnorm and knorm, this approach cannot be naturally extended to large-scale practice (for example, 30B-A3B models). Therefore, we decided not to pursue this comparison further at an early stage.
>
>
> # Response to Q1: Computational and Time Cost of CPT and Recovery
>
> For reference, our training framework supports the A800 GPU. Continued pretraining (CPT) for 22B tokens requires approximately 600 GPU-hours, while 1T tokens would require around 30,000 GPU-hours.
>
>
> # Response to Q2: Benefits at Larger Scale
>
> Based on our preliminary exploratory experiments, both the 1B and 3B models exhibited clear advantages over GQA. Consequently, we conducted more extensive and complete training on the 1B model, primarily due to cost considerations, and then further scaled up our experiments to the 30B-A3B setting. In all cases, the performance remained satisfactory. We have not found any evidence that increasing the model size diminishes these gains. On the contrary, the benefits of our approach appear to be consistent, and may even become more pronounced as the model scale increases.

---

> > ### Comment · Reviewer_mviB · 2025-11-26
> >
> > While I still view the downstream performance gains as modest, the rebuttal has clarified the optimization stability provided by the specific combination of HLC and GroupNorm. Furthermore, after reconsidering the paper's engineering contribution and taking into account the positive assessments from Reviewer GWxh and Reviewer ENbu, I am convinced that the practical value of the proposed method outweighs my initial concerns about novelty.
> >
> > Therefore, I have decided to raise my score to 6 and the Soundness score to 3.

---

### Official Review · Reviewer_GWxh · 2025-11-03

**Soundness:** 3
**Presentation:** 3
**Contribution:** 2
**Rating:** 6
**Confidence:** 3

**Summary:**

This paper advocates for interaction between heads within attention in order to improve the accuracy of transformers.

It formulates a mathematical framework that could be specialized to define different types of attention (MultiHead Attention (MHA), Grouped Query Attention (GQA), Differential Attention (DA), and Talking Heads Attention (THA)) that group or interact attention heads in different ways, then generalizes this framework to propose MultiHead Explicit Attention (MEA).

The proposed MultiHead Explicit Attention also enables compression of keys and values with limited drop in accuracy after continual pretraining.

**Strengths:**

- Clear background / motivation / formulation, generalizing different types of attention: MQA and MHA, DFA and THA
- Provides intution when and why each type of attention failes or degenerates into MHA
- Evaluated on a wide range of challenging Math and resoning benchmarks

**Weaknesses:**

- The paper did not compare with other approaches that save on KV cache by continuous pretraining.
- It wasn't clear to me why the proposed approach is more robust to KV compression than other approaches.

**Questions:**

- It might make more sense to make Section 3.2 Talking-Heads Attention (TFA) come before Section 3.1 Differential Transformer (DFA) subsection, as authors explain in the DFA subsection that it is a special case of TFA.
- Equation 10: Please also define using mathematical summation expression
- Line 203: In previous sub-section was d used to represent full embedding dimension?
- Table 2: What does "Recov." mean? Was it explained in the paper body?
- Table 2:
   - Please add a column to show compute or memory savings of each approach.
   - To make sense of the numbers, there is a need to compare with performing continual pretraining on the same dataset using other attention compression techniques in literature tuned to save the same amount of compute/memory as the approaches presented in Table 2. i.e., for the same savings of KV cache and for the same continual pretraining budget, we would like to compare the propose approach with other approaches.
   - Also, it would be better to compare evaluations on long context benchmarks after KV compression

These suggestions are long-term recommendations and may not be feasible for the rebuttal:
- For best practices of hyperparameter transfer I recommend looking at:
  - muP: https://www.cerebras.ai/blog/the-practitioners-guide-to-the-maximal-update-parameterization
  - CompleteP: https://arxiv.org/abs/2505.01618
- FYI, There are papers that describe the early phases of training as reducing bias (i.e.,  trying to minimize average loss for most samples), and later phases of training as reducing variance (i.e., handling corner cases among training samples). This paper explains it well: https://arxiv.org/abs/2502.15938

---

> ### Author Response · Authors · 2025-11-25
> **Response to Reviewer GWxh**
>
> # Response to W1: Lack of Comparison with Other Continuous Pretraining KV Cache Approaches
>
> Our work primarily focuses on the design of attention mechanisms, which naturally leads to an effective KV cache compression scheme. As previously discussed, our approach results in almost no performance degradation on non-math tasks. We consider our method to be largely orthogonal to existing KV cache compression techniques, and are interested in exploring the integration of both approaches in future work. This, however, does not diminish the contribution of our current study.
>
> Additionally, it is worth noting that recent methods for reducing KV-cache size—for example, Reducing Transformer Key-Value Cache Size with Cross-Layer Attention (NeurIPS 2025, https://openreview.net/forum?id=M2UzLRoqic)—are typically evaluated under very aggressive model configurations (such as 64 attention heads for 1B/3B models) and on relatively simple tasks like OBQA, where substantial performance drops have already been observed.
>
> In fact, we have also experimented with the MHA2MLA approach (https://arxiv.org/abs/2502.14837). From the perspective of equivalent computational cost, its scalability is suboptimal. Additionally, due to the design of Qwen models, such as qnorm and knorm, this approach cannot be naturally extended to large-scale practice (for example, 30B-A3B models). Therefore, we decided not to pursue this comparison further at an early stage.
>
> # Response to W2: Robustness to KV Compression
>
> The robustness of our approach to KV compression primarily arises from the underlying attention mechanism design. Our architecture allows for efficient information flow even when the KV cache is substantially compressed, as evidenced by the consistently strong performance across a range of tasks. This level of robustness is challenging to achieve for most alternative methods, particularly at high compression ratios.
>
>
> # Response to Q1: Section Ordering of TFA and DFA
>
> We appreciate the suggestion. We have considered presenting Section 3.2 (Talking-Heads Attention, TFA) before Section 3.1 (Differential Transformer, DFA), since DFA is a special case of TFA as explained in the manuscript. Our current structure was intended to follow a "special case–core mechanism–generalization" sequence for clarity. We will take this feedback into account and may revise the section order in future versions.
>
>
> # Response to Q2: Definition of Equation 10
>
> The "einsum" notation, widely used in the matrix computation community, was adopted for its clarity in expressing computation flows. We hope this makes the underlying process more transparent to readers:
>
> $$
> \text{HLC}(\mathbf{W}^\top, \mathbf{T})_{i,j,k} := \sum_{j=1}^{h}\sum_{j^\prime=1}^{h^\prime}\mathbf{W}_{j', j}\cdot \mathbf{T}_{i,j^\prime,k}
> $$
>
> I am not entirely sure whether this formula can be rendered correctly here. If needed, please feel free to copy it into any LaTeX-compatible editor or software to check the rendering.
>
>
> # Response to Q3: Usage of "d" in Previous Sub-section
>
> In the main text, we do not use "d" to denote the full embedding dimension. In Appendix A, "d" is used to represent the head index, which may cause confusion. We will address this issue and replace the notation in future revisions.
>
> # Response to Q4: Clarification of "Recov." in Table 2
>
> As noted in lines 456–458, even under full compression, the model can recover to a competitive performance level when continued pretraining (CPT) is combined with an additional recovery stage. The specific data used for this process is further detailed in Appendix C.
>
> # Response to Q5: Compute or Memory Savings Table
>
> Certainly. Below is a table summarizing the remaining KV ratio for each compression level:
>
> | Model Compression Stage | Remained KV Ratio |
> |------------------------|-------------------|
> | Half Compression       | 75%               |
> | Deep Compression       | 53.125%           |
> | Full Compression       | 50%               |
>
> ## On the Lack of Traditional Long-Context Benchmark Evaluation
>
> The tasks used in our experiments are not only highly challenging but also inherently long, with average chain-of-thought lengths exceeding 18k tokens. Due to these characteristics, we did not explicitly evaluate the models on traditional long-context benchmarks in this study. However, we fully acknowledge the value of such evaluations and are planning additional experiments to address this point. Once new results are obtained, we will share them at the earliest opportunity.
>
> # Reference to Phases of Training: Bias and Variance
>
> Thank you for pointing out this perspective and sharing the reference. We appreciate the suggestion and will consider discussing this aspect in future revisions.

---

### Official Review · Reviewer_ENbu · 2025-11-11

**Soundness:** 3
**Presentation:** 3
**Contribution:** 3
**Rating:** 6
**Confidence:** 3

**Summary:**

Paper proposes an explicit Multi-head Explicit Attention method that enables cross head interaction. It is a generalization of prior methods like Differential Transformers and Talking Heads Attention, using a Head-level Linear Combination unit. While the formulation is interesting, overall the performance is not much different than the state-of the art baseline Differential Transformers. Furthermore, evaluation is weak, only applied to one type&size of model and does not include any long context evaluations.

**Strengths:**

- Address an important bottleneck in LLMs which is attention for long contexts.
- Formulation of generalization of prior inter head interaction methods like Differential Transformers and Talking Heads Attention is interesting.

**Weaknesses:**

- While the KV cache compression is targeting bottleneck for long contexts, there is no long context evaluation in the paper.
- The full-parameter CPT setup reduces the performance of math benchmark baseline. This opens up the question whether math is more sensitive to compression or is it because CPT dataset needs to have more math data  in it. It's hard to tell the reason from the data.
- There is no comparison with other kv cache compression methods.
- There is no validation for the learning rate selection method which is based on curve fitting.
- The performance of the proposed method is not much better than the Differential Transformer baseline.

**Questions:**

- Why does CPT applied on top of the Baseline Qwen3 models reduces the performance of math task? Is it because math is more sensitive to compression or is it because CPT dataset needs to have more math data  in it? Or both? It's hard to tell the reason from the data.
- Can you add long context evaluation benchmarks such as RULER, NIH?
- How did you validate the learning rate selection method?
- Is the code going to be open source?

---

> ### Author Response · Authors · 2025-11-25
> **Response to Reviewer ENbu**
>
> # Response to W1 and Q2: Lack of Long-Context Evaluation
>
> Although KV cache compression is often viewed as a solution for bottlenecks in long-context scenarios, its benefits are not limited to this case alone. In our work, we find that KV cache also presents a bottleneck when increasing model inference concurrency, making its compression broadly valuable. In addition, the tasks used in our experiments are not only highly challenging but also inherently long, with average chain-of-thought lengths exceeding 18k tokens. Because of these characteristics, we did not explicitly evaluate on traditional long-context benchmarks in this study. However, we fully acknowledge the importance of such evaluations, and we are planning further experiments to address this point. Once new results are obtained, we will share them at the earliest opportunity.
>
>
> # Response to W2 and Q1: Math Benchmark Performance and CPT Setup
>
> We appreciate these thoughtful questions and fully agree that understanding the source of math performance changes under CPT is important. First, we would like to clarify that the full-parameter CPT setup does not involve any model pruning, so any observed drop in math performance should be attributed to the data itself, rather than to compression effects. This naturally raises some interesting possibilities: math tasks may be inherently more sensitive to shifts in data composition, or it could be that further increasing the proportion of math-related data in the CPT corpus would help mitigate these effects.
>
> Importantly, our observations indicate that the overall quality of our CPT data remains high. It is also relevant to note that our starting point, Qwen3-30B-A3B, is an instruct-tuned model, which means it has already undergone SFT and RL. As a result, the CPT process could, in principle, lead to some degradation of model capabilities. However, in practice, we find that applying CPT does not cause performance degradation on other tasks. This reassures us that the CPT process itself is not generally harmful to model performance. Accordingly, we believe that using the full-parameter CPT result as a baseline for pruned CPT experiments trained on the same data is reasonable.
>
> For additional context, prior work on parameter pruning (SVDLLM, ICLR 2025, https://arxiv.org/abs/2403.07378) and on reducing KV-cache size (Reducing Transformer Key-Value Cache Size with Cross-Layer Attention, NeurIPS 2025, https://openreview.net/forum?id=M2UzLRoqic) has reported noticeable performance drops even at relatively low pruning ratios, and these studies have not evaluated large-scale models on challenging reasoning tasks.
>
> Overall, we see our setup as a reliable basis for further exploration. We also agree that investigating approaches such as fine-tuning the data mix or probing the sensitivity of specific tasks could provide valuable insights for future research.
>
> # Response to W3: Comparison with Other KV Cache Compression Methods
>
> Our work primarily focuses on the design of attention mechanisms, which naturally leads to an effective KV cache compression scheme. As previously discussed, our approach results in almost no performance degradation on non-math tasks. We consider our method to be largely orthogonal to existing KV cache compression techniques, and are interested in exploring the integration of both approaches in future work. This, however, does not diminish the contribution of our current study.
> Additionally, it is worth noting that recent methods for reducing KV-cache size—for example, Reducing Transformer Key-Value Cache Size with Cross-Layer Attention (NeurIPS 2025, https://openreview.net/forum?id=M2UzLRoqic)—are typically evaluated under very aggressive model configurations (such as 64 attention heads for 1B/3B models) and on relatively simple tasks like OBQA, where substantial performance drops have already been observed.
>
> # Response to W4 and Q3: Validation of Learning Rate Selection via Curve Fitting
>
> Our experiments show that the results of the main trials, such as the observed partial order of loss values, are consistent with those from our learning rate selection process, thereby validating our approach. We further conducted curve-fitting predictions using only early-stage data points, finding that the predicted curves closely align with the actual data. Even when using only the first 10% of data points, the prediction error remains within ±5%. Notably, the ranking among learning rates stays stable throughout, further supporting our selection method.

---

> ### Author Response · Authors · 2025-11-25
> **Response to Reviewer ENbu Part 2**
>
> # Response to W5: Performance Compared to the Differential Transformer Baseline
>
> Our findings demonstrate that even in the absence of explicit denoising of attention scores, allowing attention scores to undergo free linear combination—when combined with Group Normalization—can achieve performance that is comparable to the Differential Transformer. This highlights the validity of our approach and introduces a new perspective for future research. Furthermore, as is often the case with architectural improvements, large performance gains are uncommon. For example, [NeurIPS 2025 Spotlight] TPA: Tensor ProducT ATTenTion Transformer (T6) (https://arxiv.org/abs/2501.06425).
>
>
> # Response to Q4: Open Sourcing the Code
>
> The code will be made available as open source once the paper is released.

---

> > ### Comment · Reviewer_ENbu · 2025-11-25
> >
> > Thank you for your response. Acknowledging that I've read it and have no further questions.

---

### Note · Program_Chairs · 2026-01-17
**Submission Desk Rejected by Program Chairs**

The following references in this submission do not refer to real documents and/or have major errors in bibliographic information:

 Tianle Sun et al. Livemathbench: A continuously updated benchmark for evaluating mathematical reasoning of llms. arXiv preprint arXiv:2412.04468, 2024.
Jason Wei et al. Climaqa: A benchmark for evaluating large language models on climate science. arXiv preprint arXiv:2407.05595, 2024.
Xuehai Liu et al. Olympiadbench: A benchmark for ai mathematical olympiad problems. arXiv preprint arXiv:2404.07762, 2024b.